# Self-Assembly of Molecular Landers Equipped with Functional Moieties on the Surface: A Mini Review

**DOI:** 10.3390/ijms25116277

**Published:** 2024-06-06

**Authors:** Nadia El Hasnaoui, Ahmed Fatimi, Youness Benjalal

**Affiliations:** 1Department of Chemistry, Polydisciplinary Faculty, Sultan Moulay Slimane University, P.O. Box 592, Mghila, Beni-Mellal 23000, Moroccoa.fatimi@usms.ma (A.F.); 2Chemical Science and Engineering Research Team (ERSIC), Polydisciplinary Faculty, Sultan Moulay Slimane University, P.O. Box 592, Mghila, Beni-Mellal 23000, Morocco

**Keywords:** self-assembly, molecular Landers, functional groups, scanning tunneling microscopy, van der Waals forces, hydrogen bonding, electrostatic interactions, theoretical calculations, frontier molecular orbitals

## Abstract

The bottom-up fabrication of supramolecular and self-assembly on various substrates has become an extremely relevant goal to achieve prospects in the development of nanodevices for electronic circuitry or sensors. One of the branches of this field is the self-assembly of functional molecular components driven through non-covalent interactions on the surfaces, such as van der Waals (vdW) interactions, hydrogen bonding (HB), electrostatic interactions, etc., allowing the controlled design of nanostructures that can satisfy the requirements of nanoengineering concepts. In this context, non-covalent interactions present opportunities that have been previously explored in several molecular systems adsorbed on surfaces, primarily due to their highly directional nature which facilitates the formation of well-ordered structures. Herein, we review a series of research works by combining STM (scanning tunneling microscopy) with theoretical calculations, to reveal the processes used in the area of self-assembly driven by molecule Landers equipped with functional groups on the metallic surfaces. Combining these processes is necessary for researchers to advance the self-assembly of supramolecular architectures driven by multiple non-covalent interactions on solid surfaces.

## 1. Introduction

An essential approach toward the massively bottom-up synthesis of functional nanostructures is the controlled self-assembly of molecular components adsorbed on solid substrates [1,2,3]. This active domain has been widely studied [4,5,6]. For example, organic molecules have been adsorbed on a metallic surface where metal clusters have been created spontaneously by trapping diffusing adatoms by STM manipulation of the molecular mold [7,8]. These findings suggest that it is indeed possible to build extended metallic nanostructures, such as nanowires, for potential use in emerging molecular electronic circuitry, by utilizing the molding function of organized structures formed from large organic molecules. However, the versatility of these processes is limited by the special requirements of the substrate, and self-assembly controlled through vital properties of appropriately designed Lander molecules would be advisable [9]. Intermolecular non-covalent interactions, such as the van der Waals (vdW) force, hydrogen bonding (HB), an electrostatic interaction, or metal–organic coordination, have the spatial directionality and selectivity necessary to form zero-dimensional (0D) nanodots, one-dimensional (1D) chains, and two-dimensional (2D) networks. Several studies have demonstrated that melamine (M) and cyanuric acid (CA) molecules can form stable self-assembled nanostructures on various metallic substrates like Au(111), highly oriented through triple hydrogen bonding (HB) provided by diamino-triazine (DAT) and di-carboxylic imide (DCI) functional groups [10,11,12]. Other bimolecular systems with similar complementary hydrogen bonding recognition sites can also form triple hydrogen bonds on surfaces [13,14,15,16,17,18,19]. To decouple the functional groups of the molecule from the underlying metal substrate, a family of large organic molecules, known as Landers, was synthesized and studied on surfaces using a scanning tunneling microscope (STM). This research began as early as the year 2000 [5,6,20,21], consisting of a rigid polyaromatic board with overlapping π-orbitals, which constitute the wire part, and four spacer groups as legs, which are attached to the polyaromatic planar to decouple it from the surface (Figure 1). The chains formed by these Lander molecules on terraces were observed only on special surfaces [22]. A similar structure was also detected along the $\left[11\bar{2}\right]$ direction on terraces of the metallic surfaces but was not thoroughly characterized [23]. However, there is an insufficiency of data relating to the assembly of molecular structures driven by HB on the substrates. In this respect, molecular Landers are an interesting class of complex organic compounds [24,25,26]. The new Lander molecules equipped with functional moieties, which will foster complementary hydrogen bonding as the mechanism responsible for the formation of various molecular nanostructures on the metallic surfaces were characterized by a combination of UHV-STM (ultra-high vacuum-scanning tunneling microscopy) and described through theoretical studies [9,27,28,29,30]. The self-assembly of Lander molecules, driven by non-covalent interactions (hydrogen bonding, van der Waals forces, electrostatic interactions, metal–organic coordination …), is an important point and appears to be a practical strategy since it allows the fabrication of distinct nanostructures. Thus, it has numerous applications such as metallic wires, networks, or arrays, for connecting and forming molecular quantum devices for electronic circuitry. Here, we review some representative works using a combination of theoretical studies of molecular structures, calculated STM images, and scanning tunneling microscope (STM) imaging to investigate the deposition and co-deposition of Lander molecules equipped with functional moieties on the metallic substrate. These works provide key findings from STM and calculations on surface-adsorbed molecular Landers and outline a set of processes that will be adopted in future works for the self-assembly of Lander molecules on metallic substrates to form distinct networks. 

## 2. Adsorption of a Single Lander Molecule on Metal Surfaces

In recent years, new classes of Lander molecules equipped with functional groups have garnered increased interest in metallic substrate adsorption, including altered diffusion behavior, controlled electronic coupling to step edges, and are used as molds to create metallic nanostructures at terraces and step edges [7,31,32,33,34]. In addition, molecular Landers have been assembled into wires, employing substrate template effects [35], and into the organization of networks by non-covalent interactions, for example, hydrogen bonding (HB), which is enabled by the functional groups, and van der Waals (vdW) interactions [27,29,30]. In 2009, Miao Yu et al. [9] characterized a newly Lander-DAT molecule, $C_{64}H_{68}N_{10}$, by combining imaging and calculated STM images; it consists of two DAT functional groups that enable intermolecular HB interactions N−H⋯N, and a hexaphenylbenzene core with four spacer legs (tert-butyl), as shown in Figure 2a. When Lander-DAT molecules are sublimated on the metallic surfaces, distinct isolated molecules are shown on the terraces. The optimum adsorption position for a Lander-DAT molecule is identified with the molecular mechanics MM4 code [36], where all the atoms closest to the metallic surfaces (Au(111) and Cu(110)) are at a height of 2.50 Å, their board is parallel to the terrace of the surface, and the two DAT functional moieties are not parallel to the substrate (Figure 2d,e). Moreover, the leg groups caused the hexaphenylbenzene core to be lifted to a height of 4.50 Å above the substrate, which allowed the platform to be isolated from the surface. The minimal adsorption energy between the Lander-DAT and the metallic surface is usually small, suggesting that the Lander molecule is physisorbed. Theoretical simulations of the STM images of an individual Lander-DAT on Au(111) and Cu(110) surfaces were obtained with the EHMO-ESQC code (extended Hückel molecular orbital theory–elastic scattering quantum chemistry) [37]. This technique, based on the EHMO method, is used to study the transmission of electrons through a defect embedded in an infinite system. In 1988, the method was used to study the transmission of electrons through a molecule embedded in a conducting polymer [38,39,40]. More recently, the method was further developed to study the tunneling of electrons in a scanning tunneling microscope (STM), which consists of the apex of the molecule being imaged and the substrate where the molecule is adsorbed [41]. The results of STM and ESQC simulations for Lander-DAT show four bright lobes in a rectangular shape, corresponding to the legs ($C_{4}H_{9}$) of the molecule, while the sub-protrusions have been attributed to the hexaphenylbenzene in the center (Figure 2b,c,f,g). Interestingly, no features can be attributed to the DAT moieties in the ESQC and STM images. This confirms the ability of the STM measurements and theoretical calculations to characterize the morphologic and topographic properties of the Lander molecules adsorbed and substrate–molecule interactions, which are responsible for the self-assembly of network nanostructures on the substrate. In addition, based on the manipulation experiments and calculation results for the Lander-DAT molecules on the surfaces, researchers found that the molecular mobility and diffusion barrier for Lander-DAT are much higher on Cu(110) than on Au(111) because the close-packed substrate is less corrugated than that of the more open substrate. Based on these findings, it can be concluded that the Au(111) substrate may be the more suitable surface for forming ordered nanostructures of molecular Landers driven by intermolecular interactions.

In 2010, Miao Yu et al. [29] investigated a new Lander-DCI molecule, $C_{112}H_{102}N_{2}O_{4}$, forming a central molecular board, four DTP (bulky 3,5-di-tert-butylphenyl) moieties as spacer legs, and two DCI (di-carboxylic imide) functional moieties, to enable the N−H⋯O intermolecular HB on the substrate, as shown in Figure 3a, using extended Hückel molecular orbital theory–elastic scattering quantum chemistry (EHMO-ESQC) [37,38,39,40,41] and the molecular mechanics MM4(2003) code [36] to relax the molecule on the substrate, demonstrates that the STM and calculated ESQC images of Lander-DCI on Au(111) are formed by four bright protrusions attributed to the four DTP legs; thus, the central molecular board and the DCI moieties are not visible in the ESQC and STM images.

To describe why the experimental STM and calculated ESQC images of these Lander molecules show that the functional groups are not visible, we calculated the frontier molecular orbitals (FMOs) of Lander-DAT and Lander-DCI alone (Figure 4 and Figure 5) by using the EHMO method as implemented in the YAeHMOP code [42,43,44,45]. These calculations show that the electron density of the frontier molecular orbitals of DCI and DAT Landers is delocalized across the molecule. However, for the LUMO orbital, it is higher on both functional groups (DAT and DCI), the core, and shows an average electron density over the spacer groups (legs). Conversely, the HOMO orbitals are characterized by a very high electron density on the legs (tert-butyl and bulky 3,5-di-tert-butylphenyl) and very low density on the functional moieties for both molecules. In addition, the Lander molecules are physisorbed on the metallic surfaces, and the molecular board of each molecule Lander is elevated from the surface by the legs. Therefore, the surface does not influence the electronic structures of the molecules when they are adsorbed. From these results, we demonstrate that the morphology of the STM image is influenced by the contribution of the HOMO molecular orbital in the measurements and calculated images of both Landers because the bias voltage (V) used in all cases is lower than the HOMO-LUMO gap of the DAT and DCI molecules (of approximately, 2.65 V for Lander-DAT and 2.05 V for Lander-DCI) [9,29,46,47].

## 3. Self-Assembly Formed by Lander Molecules Guided by vdW and HB Interactions

Self-assembly experiments of Lander-DAT were performed by the Miao Yu group [29]; distinct 2D structures were observed on the Au(111) substrate, such as ‘Four-Blade Mill’, ‘Transition’, and ‘Stripe’, as illustrated in Figure 6. They investigated these structures by a combination of STM imaging and molecular mechanics modeling with the MM4 force field [36], as well as calculated STM images (ESQC) [37,38,39,40,41]. A key point accounting for the illustrated molecular Lander-DAT nanostructures is the presence and the conformational flexibility of DAT functional moieties, which are further linked to the hexaphenylbenzene rings of the molecular board by the σ link and decoupled from the metallic substrate by the legs (tert-butyl). The calculated results show an intermolecular network optimized in 3D by N−H⋯N HB, in the range of 2.8–3.0 Å, between DAT functional groups, in addition, the average corresponding energy is 0.82 eV, including HB and vdW interactions. To explain the role of the DAT moieties in these described structures, researchers have synthesized and characterized a related compound, Lander-ND, $C_{58}H_{62}$, which has an identical structure to Lander-DAT, but without the DAT moieties [27]. The calculated and experimental images of Lander-ND on the Au(111) surface show a morphology similar to the STM and ESQC images of Lander-DAT. However, for the 2D network, only a single network has been observed on the metallic surface. Therefore, the DAT functional groups drive the self-assembly and the formation of distinct supramolecular nanostructures on the metallic substrate through intermolecular HB and vdW interactions.

In 2010, Miao Yu et al. announced their work [29], where they investigated and analyzed the self-assembly of Lander-DCI on an Au(111) substrate. At low coverages, extended 1D molecular chains oriented along the $\left[11\bar{2}\right]$ directions as presented in Figure 7a–c. These results show that the chains consist of Lander-DCI molecules aligned along the chain direction and interconnected through double N−H⋯O HB with an average bond length of 2.2 Å and a relatively high stabilization energy of 0.5 eV. Figure 7f shows an extended 2D molecular form at high resolution on the Au(111) substrate, consisting of 1D chains formed by Lander-DCI molecules aligned head-to-tail via double N−H⋯O intermolecular HBs, while van der Waals interactions link them together.

An interesting potential application is to form self-assembled heteromolecular networks through the co-adsorption of different Lander molecules. This concept has been demonstrated in various biomolecular systems that do not have elevating legs [11,12,15,48]. N. Kalashnyk, et al. introduced this process in 2014 [30] by studying the co-adsorption of two Lander molecules, DAT and DCI, on an Au(111) surface. This work demonstrates that on the terrace of the Au(111), the co-deposition of Landers DAT and DCI results in the formation of 2D molecular nanostructures, as depicted in Figure 8. They found that the Lander-DCI molecules are aligned in chains via head-to-tail interactions and stabilized by double N−H⋯O HBs (2.51 Å) identical to chains formed by Lander-DCI alone, while Lander-DAT molecules separate these chains by two N−H⋯O HBs (2.66 Å and 3.46 Å), between the DAT group and DCI group in Lander-DCI molecules (solid lines in Figure 8c). The total energy per DAT–DCI molecular pair is higher than the total energy of DAT-DAT and DCI-DCI pairs alone (1.51 eV vs. 0.50 eV for the 2D network DCI-DCI and 0.82 eV for the 2D network DAT-DAT). These results confirm that the 2D heteromolecular network, formed by Lander-DAT and Lander-DCI, is energetically more favorable than networks formed by either Lander-DCI or Lander-DAT alone.

## 4. Self-Assembly Formed by Lander Molecules Guided by vdW, HB, and Electrostatic Interactions

Many works have shown that permanent charges on molecules with ionic character generate electronic interactions that influence the formation of molecular structures on surfaces with alternate anions and cations [5,49,50,51]. In 2012, the Miao Yu group characterized the self-assembly of PTCDI molecules (perylene-3,4,9,10-tetracarboxylic diimide), $C_{12}N_{2}H_{10}O_{4}$, along with Ni atoms on a metallic substrate, using a combination of STM measurements and simulated STM images with the EHMO-ESQC method [37,38,39,40,41,52]. After optimizing the systems on the substrate with the molecular mechanics MM4(2003) code [36], to establish the nanostructures on the surface, they demonstrated that it is possible to synthesize different types of nanostructures (Figure 9), such as zero-dimensional (0D) structures involving three PTCDI molecules assembled primarily by vdW interactions and weak HBs surrounding a central Ni atom (Figure 9b,f,j), one-dimensional (1D) chains of PTCDI molecules linked by double N−H⋯O HBs (3.3 Å), with each molecule surrounded by four Ni atoms near CDI groups (Figure 9a,e,i), and the two-dimensional (2D) network, as displayed in Figure 9d,h. This network is formed by 1D PTCDI chains (marked as ‘A’ in Figure 9d) aligned head-to-tail via double HB, interlinking PTCDI molecules (marked as ‘C’ and ‘B’ in Figure 9d). They showed that Ni atoms are present in nanostructures, but are not featured in STM and ESQC images due to the electronic coupling between these atoms and the CDI functional groups [53].

To maximize the stability of the deposited molecular architectures, Ni-PTCDI, the Miao Y. and Benjalal Y. groups investigated and characterized the co-adsorption of the Lander-DAT with the PTCDI molecules and Ni metal atoms on an Au(111) substrate [54] (Figure 10), by combining STM observations, molecular mechanics modeling (MM) [36], and ab initio calculations with the Mopac2009 code using the SCF-MP6 level [55]. They found that the average bond length of N−H⋯O hydrogen bonding between neighboring planar PTCDI molecules is 2.45 Å and the distances between Ni and O atoms, of the nearest CDI groups, are between 3.75 Å and 4.20 Å, which are much too large, so the coordination bonding of Ni-O is not possible [56,57,58,59]. Each Ni atom acquires a negative partial charge (−0.50|e|); the average net charges for the atoms close to the nickel atom are as follows: −0.60|e| for the nitrogen atoms in the DAT moieties, −0.60|e| and −0.45|e|, respectively, for the oxygen and nitrogen atoms in the CDI functional groups in the PTCDI molecule, and +0.57|e| for the carbon atoms in both molecules. In this case, they found that the Ni atoms maintain the 2D network through electrostatic interactions between the nickel atoms and the functional moieties. This attractive interaction brings the DAT groups closer to the CDI groups, facilitating 3D HB (Figure 10c,d). Furthermore, they demonstrated that this mixed assembly (DAT-PTCDI–Ni) is more energetically favorable than either homogeneous or heteromolecular networks on the metallic substrate.

## 5. Conclusions

In this mini-review, we explored the self-assembly of Lander molecules through intermolecular non-covalent interactions. These studies demonstrate that the co-adsorption of Lander molecules, elevated above a surface by legs and bearing complementary functional groups, can form distinct, stable self-assemblies via hydrogen bonding (HB), van der Waals (vdW) forces, and electrostatic interactions. Moreover, the 3D hydrogen bonding between the functional groups of the Lander molecules in heteromolecular networks, facilitated by electrostatic attraction, is significantly stronger than the interactions between the Lander molecules alone. The processes adopted here for the formation of stable structures through 3D-optimized hydrogen bonding are linked to the freedom and conformational flexibility of the functional moieties of molecular Landers. It may be possible to synthesize and characterize new Lander molecules on various substrates to advance the self-assembly of Lander molecules, directed by non-covalent interactions, for forming distinct networks, such as arrays of nanodots, metallic wires, or networks, which are useful in the field of nanodevices and nanotechnology. 

## Figures and Tables

**Figure 1 ijms-25-06277-f001:**
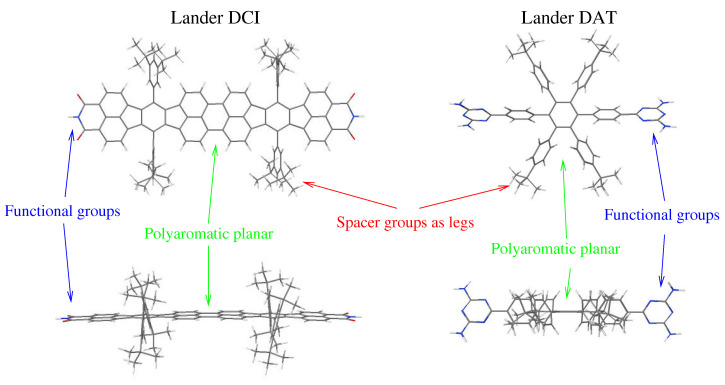
Chemical structures of Lander-DCI and Lander-DAT, forming a polyaromatic hydrocarbon equipped with four spacer legs, and two functional groups, DAT and DCI.

**Figure 2 ijms-25-06277-f002:**
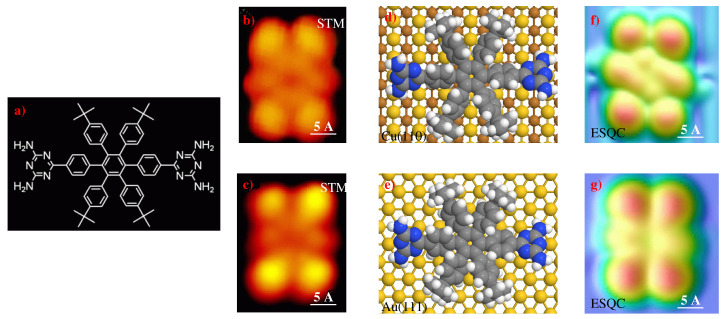
(**a**) Chemical structure of molecular Lander-DAT ($C_{64}H_{68}N_{10}$). (**b**,**c**) STM images of an individual Lander-DAT on Cu(110) and Au(111) surfaces, respectively ($I_{t}=-0.66$ nA; $V_{t}=-1.73$ mV). (**d**,**e**) space-filling models of an individual Lander-DAT adsorbed on Cu(110) and Au(111) respectively, where N, H, C, and surface atoms are colored in blue, white, gray, and yellow, respectively. (**f**,**g**) EHMO-ESQC calculated images of a Lander-DAT adsorbed on Cu(110) and Au(111), respectively. ((**b**,**f**) are adapted and reprinted with permission from [9]. Copyright 2010 American Chemical Society; (**a**,**c**,**g**) are adapted and reprinted with permission from [27]. Copyright 2009 Tsinghua University Press and Springer-Verlag Berlin Heidelberg).

**Figure 3 ijms-25-06277-f003:**
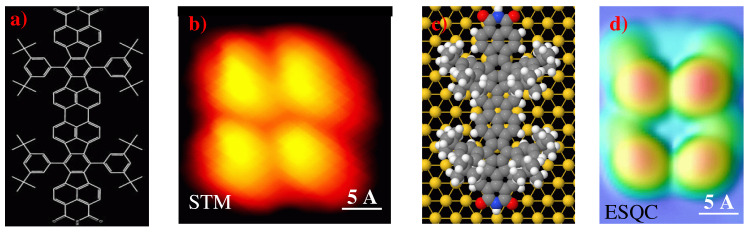
(**a**) Chemical structure of molecular Lander-DCI ($C_{112}H_{102}N_{2}O_{4}$). (**b**) STM image of an individual Lander-DCI on Au(111) substrate ($I_{t}=0.26$ nA; $V_{t}=1239$ mV). (**c**) Top view of the optimized chemical structure of a Lander-DCI on Au(111), where Au, Ni, H, O, N, and C atoms are in yellow, green, white, red, blue, and gray, respectively. (**d**) ESQC image of a Lander-DCI on Au(111) as shown in panel (**c**); ((**a**–**d**) are adapted and reprinted with permission from [29]. Copyright 2010, The Royal Society of Chemistry).

**Figure 4 ijms-25-06277-f004:**
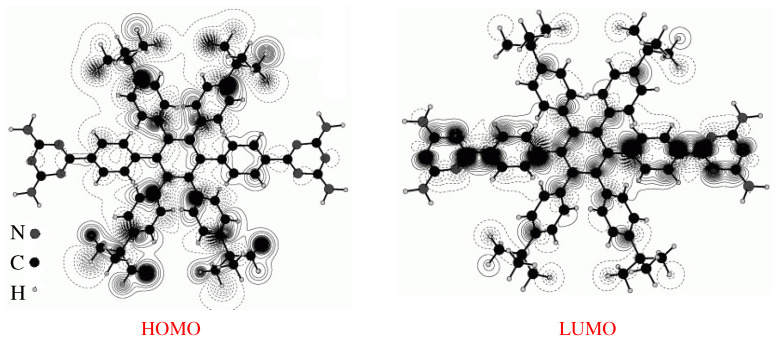
Calculated frontier molecular orbitals, HOMO and LUMO, of Lander-DAT molecule free, where N, H, and C, atoms are colored in gray, light gray, and black, respectively.

**Figure 5 ijms-25-06277-f005:**
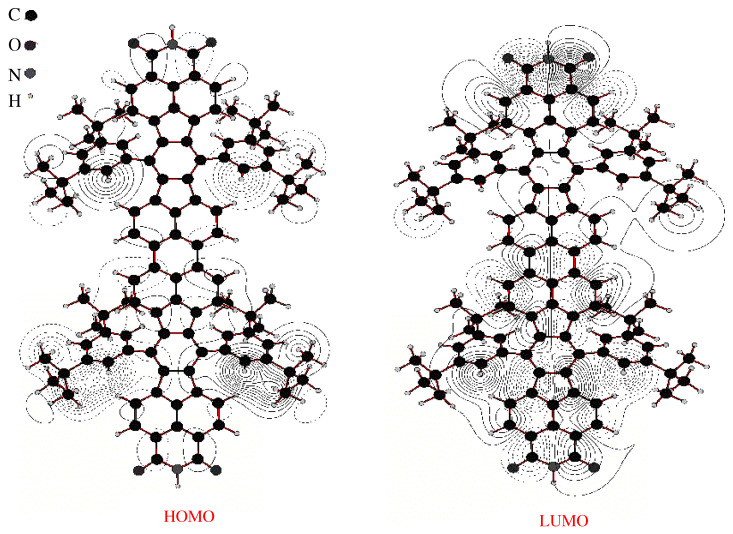
Calculated frontier molecular orbitals, HOMO and LUMO, of Lander-DCI molecule free, where N, H, C, and O atoms are colored in gray, light gray, black, and dark gray, respectively.

**Figure 6 ijms-25-06277-f006:**
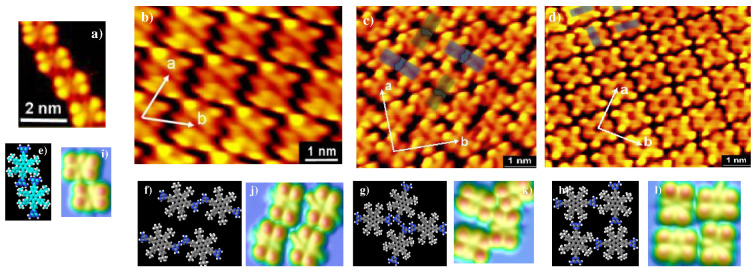
(**a**) STM image of the 1D chain of Lander-DAT on Au(111) surface ($I_{t}=0.70$ nA; $V_{t}=1250$ mV). (**b**–**d**) STM images of the ‘Stripe’ structure, ‘Four-Blade Mill’ structure, and ‘Transition’ structure, respectively, on Au(111) ($I_{t}=0.48$ nA; $V_{t}=1250$ mV). (**e**–**h**) Optimized models for the 1D chain, ‘Stripe’ structure, ‘Four-Blade Mill’ structure, and ‘Transition’ structure, respectively, where atoms of C, N, and H are in gray, blue, and white. (**i**–**l**) EHMO-ESQC calculated images of the 1D chain (**i**), ‘Stripe’ structure (**j**), ‘Four-Blade Mill’ structure (**k**), and ‘Transition’ structure (**l**) as in experiments (**a**–**d**). ((**a**–**e**) are adapted and reprinted with permission from [27]. Copyright 2010 American Chemical Society).

**Figure 7 ijms-25-06277-f007:**
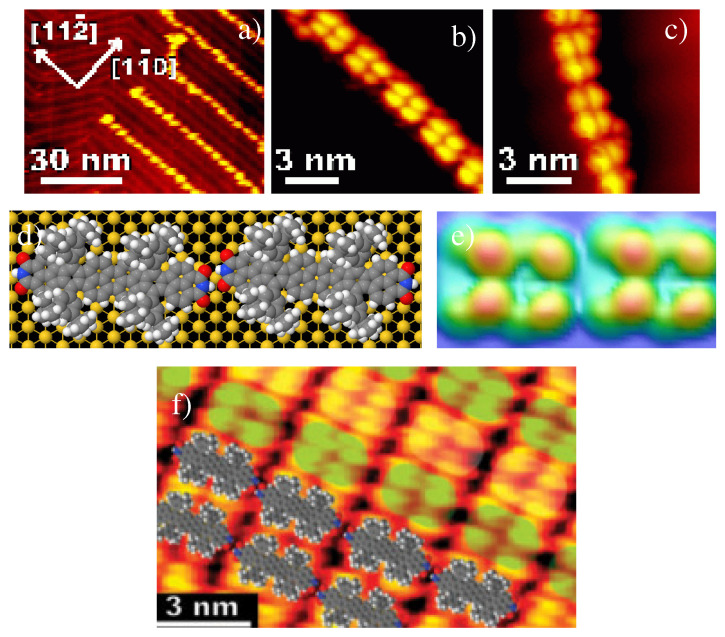
(**a**) STM image of the 1D structures of Lander-DCI on Au(111) terrace ($I_{t}=0.55$ nA; $V_{t}=1250$ mV). (**b**,**c**) STM images of a 1D chain of Lander-DCI on a terrace and a step edge of the Au(111), respectively. (**d**) Model of the 1D chain of Lander-DCI on Au(111), obtained from MM4 code. (**e**) EHMO-ESQC calculated image of a 1D chain of Lander-DCI on Au(111) as shown in panel (**d**). (**f**) STM image of the 2D network of Lander-DCI and the superimposed calculated model. ((**a**–**d**,**f**) are adapted and reprinted with permission from [29]. Copyright 2010 The Royal Society of Chemistry).

**Figure 8 ijms-25-06277-f008:**
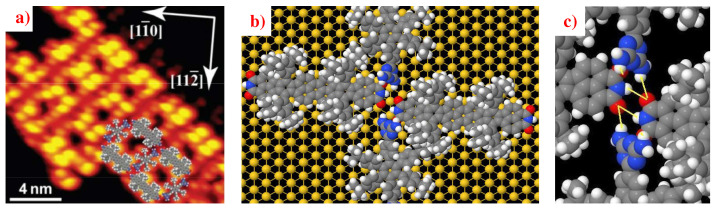
(**a**) STM image of the 2D assembly of DAT-DCI molecules on the Au(111) substrate and the superimposed calculated model. (**b**) Calculated model of the structure formed by Lander-DAT and Lander-DCI on Au(111), where Au, Ni, H, O, N, and C atoms are in yellow, green, white, red, blue, and gray, respectively. (**c**) Close-view of the 3D HB hydrogen bonding between DCI and DAT groups. ((**a**,**b**) are adapted and reprinted with permission from [30]. Copyright 2014 The Royal Society of Chemistry).

**Figure 9 ijms-25-06277-f009:**
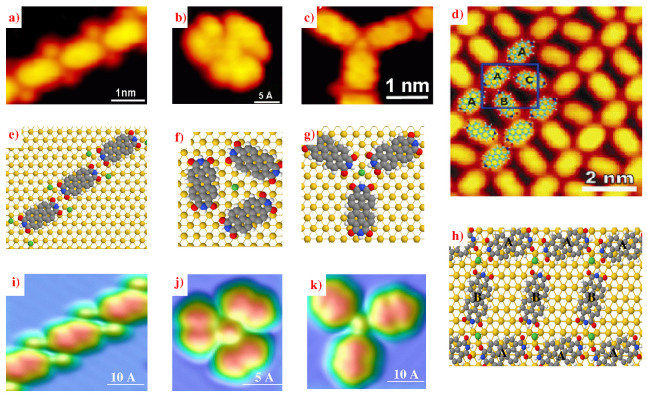
(**a**) STM image of the 1D assembly of PTCDI molecules with Ni atoms on Au(111) ($I_{t}=0.53$ nA; $V_{t}=1.20$ V). (**b**) STM image of ‘0D’ nanostructure ($I_{t}=0.55$ nA; $V_{t}=1.05$ V). (**c**) STM image of ‘Y-shape’ nanostructure ($I_{t}=0.44$ nA; $V_{t}=0.71$ V). (**d**) STM image of the 2D network of PTCDI-Ni (It = 0.50 nA; Vt = 1.75 V). (**e**–**h**) Calculated structures for the 1D chain, ‘0D’, ‘Y-shape’ nanostructures, and the 2D network, where Au, Ni, H, O, N, and C atoms are in yellow, green, white, red, blue, and gray, respectively. (**i**–**k**) EHMO-ESQC calculated images of the 1D chain (**i**), ‘0D’ cluster (**j**), and ‘Y-shape’ nanostructure (**k**); (**a**–**h**) are adapted and reprinted with permission from ref. [52]. Copyright 2012 Tsinghua University Press and Springer-Verlag Berlin Heidelberg).

**Figure 10 ijms-25-06277-f010:**
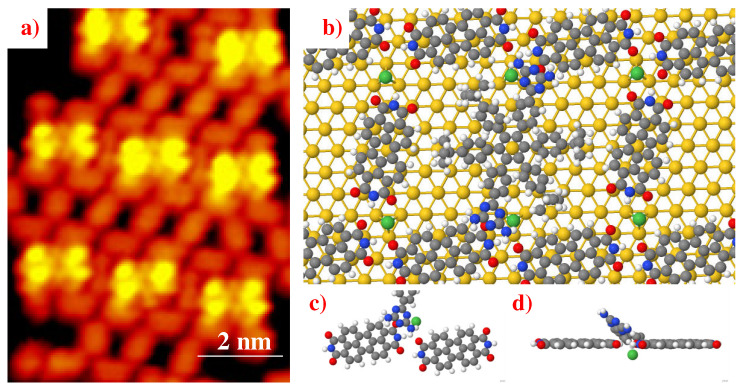
(**a**) STM image of the 2D nanostructure of Lander-DAT and PTCDI/Ni on Au(111) surface. (**b**) Calculated model of the 2D nanostructure of Lander-DAT and PTCDI/Ni on Au(111) surface, where Au, Ni, H, O, N, and C atoms are in yellow, green, white, red, blue, and gray, respectively. (**c**,**d**) Close-view and Top-view of the 3D HB between CDI, in PTCDI molecule, and DAT functional groups; ((**a**–**d**) are adapted and reprinted with permission from [54]. Copyright 2018 The Royal Society of Chemistry).

## Data Availability

The input/output files for the ESQC, MM4, Mopac, and YAeHMOP codes, which were used in this study for theoretical calculations, will be made available by the authors upon request (only for free codes).

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
