# Peer review of "Self-Assembly of Molecular Landers Equipped with Functional Moieties on the Surface: A Mini Review"

_ijms, 2024, doi:10.3390/ijms25116277_

Round 1

Reviewer 1 Report

Comments and Suggestions for Authors

The authors of the manuscript entitled "Self-assembly of molecular Landers equipped with functional moieties on the surface : a review"  have written an interesting review about structure and features of supramolecules obtained by self-assembly of Lander molecules. There are not many reviews about the topic 

The manuscript is nicely written, the examples are clearly described and the references are almost appropriate and consistent with the discussion present in the manuscript, anyway what is lacking, from my perspective, is a common thread regarding the relationship between the structure of the supramolecules obtained by the self-assembly process of the Lander molecules and the applications (and the advantages and disadvantages when, for example, a metal is used instead of another one). Shortly, what kind of supramolecules, between those described in the manuscript, is more convenient for circuitry applications?

Which kind of interactions is better to exploit in order to achieve these compounds? The application of these molecules, from an economical point of view, is convenient for vast-scale applications in nanoelectronics? (2 nm chips manufacturing is a very strategical matter nowadays)  Have been these supramolecules applied in the catalysis field? These are questions whose answers could give interesting insights into the field and add importance to the manuscript.

What is a molecular Ladder? A definition should be given at the beginning. Add definition and the appropriate references and Schemes (eventually)

Line 11: self-assembly instead of “self-assembled”

(Overall Recommendation: reconsider after major revision)

Author Response

Reviewer #1

Q1 : 

The authors of the manuscript entitled "Self-assembly of molecular Landers equipped with functional moieties on the surface: a review" have written an interesting review about structure and features of supramolecules obtained by self-assembly of Lander molecules. There are not many reviews about the topic

R1 : We appreciate this referee for the highly approval and commendation on our work.

Q2 : The manuscript is nicely written, the examples are clearly described and the references are almost appropriate and consistent with the discussion present in the manuscript, anyway what is lacking, from my perspective, is a common thread regarding the relationship between the structure of the supramolecules obtained by the self-assembly process of the Lander molecules and the applications

R2 : We agree with the referre and we have added more sentences in the section 1 (Introduction) line 51-56 :

“This is an important point, since self-assembled of Lander molecules, by non-covalent interactions, appear to be a practical strategy since it allows the fabrication of distinct supramolecular nanostructures, and has numerous applications such as metallic wires, networks, or arrays, for forming and connecting the molecular electronic circuitry and quantum devices.”

Q3 : (and the advantages and disadvantages when, for example, a metal is used instead of another one).

R3 : Examples of the advantages and disadvantages when a metal is used instead of another one :

To minimize molecule/surface interactions in order to have well-ordered structures.

Find the most favorable surface for the self-assembly of Landers molecules and strengthen the cohesion between them.

To direct the formation of extended nanostructures by bottom-up self-assembly.

However, this can lead to many attempts to choose the favorite metal surface.

Q4 : Shortly, what kind of supramolecules, between those described in the manuscript, is more convenient for circuitry applications ? Which kind of interactions is better to exploit in order to achieve these compounds?

R4 : We thank the reviewer for the constructive comment. In fact, the ordered structures produced by non-covalent hydrogen bond interactions are more stable, more coherent, and more ordered compared to structures produced by Van der Waals interactions, and we can strengthen these structures by electrostatic interactions for circuitry applications.

Q5 : The application of these molecules, from an economical point of view, is convenient for vast-scale applications in nanoelectronics? (2 nm chips manufacturing is a very strategical matter nowadays)

R5 : For the fabrication of extended metallic nanowires, the direct evidence for the molding functions of individual Lander-type molecules have been demonstrated nicely. Please see Ref. 1-4 as the typical ones, where it is shown that Lander molecules can trap and assemble metal atoms into well-defined metallic nanostructures in the cavity underneath the aromatic board by the attractive interaction of the aromatic π board with metal atoms.

Q6 : Have been these supramolecules applied in the catalysis field ?

R6 : We think that, for the Lander molecules equipped with functional groups, there is not much research of these molecules on the application of self-assembly in the catalysis field and/or they are not thoroughly characterized.

Q7 : These are questions whose answers could give interesting insights into the field and add importance to the manuscript. What is a molecular Ladder? A definition should be given at the beginning. Add definition and the appropriate references and Schemes (eventually)

R7 : We thank the reviewer for the crucial comment. To address theses questions more clearly, we have added more sentences in the section 1 (Introduction) lines 35-40 :  “To decouple the functional part of the molecule from the underlying metal surface, as early as 2000, the family of large organic molecules was synthesized and studied on surfaces by STM, these molecules namely Landers [4,18-20], consisting of a rigid polyaromatic board with overlapping π-orbitals, which constitute the wire part, and four spacer groups as legs, which are attached to the polyaromatic planar to decouple it from the metallic surface (Figure 1)”, Figure 1 and  we have cited Ref. 18-19.

Q8 : Line 11: self-assembly instead of “self-assembled”

We thank the reviewer again for the great effort and have made the revisions accordingly in the revised manuscript.

end of point-to-point response.

With best regard.

Reviewer 2 Report

Comments and Suggestions for Authors

The present submission by Benjalal and coworkers is a review article. The article touches on the topic of "molecular landers", which generally refers to organic molecules that are deposited on surfaces (ideally single monolayer). These organic molecules interact with one another in a non-covalent way, forming new assemblies / functional surfaces.

These type of structures are typically studied using STM. Periodic DFT calculations are also helpful in plotting the electron density and simulating STM-like images that can be compared to the experiment. Dr. Benjalal has been collaborating with experimental groups and has provided support on the simulation part. As I understand, the present study is supposed to be an overview to cover selected studies on the aspect and provide an insight on how STM is being combined with theory for a better understanding of the chemistry at the interface.

Over the past two decades the number of molecules deposited on surfaces and described using computational methods (including simulated STM) is on the scale of thousands. Molecules are being deposited for various reasons (e.g. qubits, memristors, catalysis etc). Many of them may be isolated or arrange themselves in some new orientation with respect to the surface. In this regard, the potential scope of the review would be significantly larger. In the present case, the authors have provided a review article with barely 40 citations. Although MDPI may not have strictly specified the number of citations involved with a typical review article, a "Minireview" in chemistry, as published by Wiley, is typically 5000 words, including around 100 references. The same is true for Nature Chemistry Reviews. 

In some cases, I can understand that originally, the authors have not clearly explained how their article is being positioned in the overall literature. However, after carefully re-reading the article, I see issues with its depth. For theoretical work, the authors would be expected to explain the molecular aspects of the surface chemistry or the overall theoretical methodologies. A current perspective by Duan and Xu shows how this is properly done https://doi.org/10.1021/acs.jpclett.3c01603. Although their article is not extensive (60 references) , it provides depth in understanding the role of the tip and substrate on the accuracy of the calcalculation. In the present submission, the authors do not go much in depth. There is no explanation of the molecular structure and properties of the involved molecules, no discussion on the computational methods, their limits and advantages for addressing the particular problem. 

In addition to the overall formatting, the 55% iThenticate report is highly concerning. It appears as if the authors did very little in terms of their own reviewing (many phrases are completely borrowed from abstracts/conclusion statements of published papers). 

Owing to these major reasons, I cannot recommend the article for publication in the present case. I suggest the authors undergo a major revision. They may decide to provide an extended review of many molecules on surfaces or alternatively keep a tutorial-like review as it is being published by RSC. Some details on the tutorial-like review article are provided at the end of this evaluation.

a) Please add molecular formulas of each molecule you describe on surfaces

b) for every molecule, please discuss its electronic structure (as an isolated molecule) and show its frontier orbitals

c) Provide a section describing the typical software and alternatives for calculating the STM images

d) For isolated models, provide LDOS and band structure

e) If you aim a tutorial review, please also provide geometries and example input/output of the calculated structures / STM images. 

f) Minor usage of figures from other papers is fine, but for review, please produce your own figures and be consistent with the colouring.

--------------------------------

Further information obtained from: https://www.rsc.org/journals-books-databases/about-journals/chem-soc-rev

Tutorial Reviews

Tutorial Reviews are concise, accessible and authoritative overviews of important contemporary topics in the chemical sciences. They should appeal to advanced undergraduates, the general research chemist who is new to the field, as well as the expert. They provide a solid introduction to the development of a subject, the latest breakthrough results and their implications for the wider scientific community. Tutorial Reviews should not contain unpublished research.

Tutorial reviews must fulfil the following criteria.

Accessible

Appeal to advanced undergraduate students and beyond. Tutorial reviews are often used in advanced undergraduate and Master’s studies.

Authoritative

Provide an essential introduction to the field which will lay the foundation of knowledge in the area, followed by the most important recent advances. Authors should include throughout the article their own insights into the development of the field and its future potential.

Exciting

In particular highlight areas where there has been a significant recent advance.

Concise

There is no strict reference limit; however please include only the most important historical and recent research, referencing the major contributions (with the “and references cited therein” addition where appropriate). Tutorial Reviews are typically up to 15 journal pages in length.

Jargon-free

Specialist terms and symbols should be defined and fundamental ideas simply explained.   

Tutorial reviews should include a 'key learning points' box, containing up to five key learning points that a reader should expect to gain from reading the review. These should be provided on submission, either at the beginning of the review or as a separate document.

Comments on the Quality of English Language

Compared to the other aspects, there are no major concerns about the English language at this stage. Many sentences may appear grammatically correct, but they imply new semantics for common words.

An example is the word methodology. Both STM and DFT would be some form of developed methods (methodologies maybe if still in development). The "authors use these methodologies to reveal some new methodologies in the area of self-assembly"  (see copied sections below).

This does not make any sense, as the characterisation and description methods are used to characterise/describe the outcomes of molecular engineering but not the methods of molecular engineering per se. 

. Herein, we review a series of studies 9 utilizing combining scanning tunneling microscopy (STM) with theoretical calculations, to reveal the methodologies used in the area of self-assembled driven by molecule Landers equipped with func- 11 tional groups on the surface.   The methodologies adopted here 232 are expected to be useful for researchers to advance the self-assembly of Lander molecules 233 directed by non-covalent interactions study in the future, within the emerging field of 234 nanotechnology and nanodevices

Author Response

Reviewer #2

Comment 1 : Over the past two decades the number of molecules deposited on surfaces and described using computational methods (including simulated STM) is on the scale of thousands. Molecules are being deposited for various reasons (e.g. qubits, memristors, catalysis etc). Many of them may be isolated or arrange themselves in some new orientation with respect to the surface. In this regard, the potential scope of the review would be significantly larger. In the present case, the authors have provided a review article with barely 40 citations. Although MDPI may not have strictly specified the number of citations involved with a typical review article, a "Minireview" in chemistry, as published by Wiley, is typically 5000 words, including around 100 references. The same is true for Nature Chemistry Reviews.

Answer 1: As for the H-bonding and electrostatic interactions directed supramolecular nanostructures, there are too many publications to be cited all.

Even so, we still add a few references as requested by this referee.

Gourdon, A. Synthesis of “molecular Landers”. European journal of organic chemistry 1998, 12, 2797–2801.

Rosei, F.; Schunack, M.; Naitoh, Y.; Jiang, P.; Gourdon, A.; Laegsgaard, E.; Stensgaard, I.; Joachim, C.; Besenbacher, F. Properties of large organic molecules on metal surfaces. Progress in Surface Science 2003 71, 95–146.

Aviram, A.; Ratner, M. A.; Molecular rectifiers. Chemical Physics Letters 1974, 29, 277-–283.

Sautet, P.; Joachim, C.; Electronic interference produced by a benzene embedded in a polyacetylene chain. Chemical Physics Letters 1988, 153, 511—516.

Sautet, P.; Joachim, C. The switching ability of a three-level tight-binding system: the isolated and embedded case. The Journal of  Chemical Physics 1988, 21, 3939-–3957.

Sautet, P.; Joachim, C. Electronic transmission coefficient for the single-impurity problem in the scattering-matrix approach. Phys. Rev. B 1988, 38, 12238—12247.

Landrum, Greg, and Wingfield Glassey. "Yet Another Extended Hückel Molecular Orbital Package (YAeHMOP) Version 3.0 User Manual." (2006).

Coratger, R., Calmettes, B., Benjalal, Y., Bouju, X., & Coudret, C. (2012). Structural and electronic properties of hexa-adamantyl-hexa-phenylbenzene molecules studied by low temperature scanning tunneling microscopy. Surface science, 606(3-4), 444-449.

Ge, X.; Kuntze, J.; Berndt, R.; Tang, H.; Gourdon, A. Tunneling spectroscopy of lander molecules on coinage metal surfaces. Chemical Physics Letters 2008, 458, 161–165.

Ruiz, E.; Alvarez, S.; Hoffmann, R.; Bernstein, J.  J. Am. Chem. Soc. 1994, 8207--8221.

Hughbanks, T.; Hoffmann, R. Chains of trans-edge-sharing molybdenum octahedra: metal-metal bonding in extended systems. Journal of the American Chemical Society 1983,  105, 3528-3537.

Comment 2 : In some cases, I can understand that originally, the authors have not clearly explained how their article is being positioned in the overall literature. However, after carefully re-reading the article, I see issues with its depth. For theoretical work, the authors would be expected to explain the molecular aspects of the surface chemistry or the overall theoretical methodologies. A current perspective by Duan and Xu shows how this is properly done https://doi.org/10.1021/acs.jpclett.3c01603. Although their article is not extensive (60 references) , it provides depth in understanding the role of the tip and substrate on the accuracy of the calcalculation. In the present submission, the authors do not go much in depth. There is no explanation of the molecular structure and properties of the involved molecules, no discussion on the computational methods, their limits and advantages for addressing the particular problem.

In addition to the overall formatting, the 55% iThenticate report is highly concerning. It appears as if the authors did very little in terms of their own reviewing (many phrases are completely borrowed from abstracts/conclusion statements of published papers).

Owing to these major reasons, I cannot recommend the article for publication in the present case. I suggest the authors undergo a major revision. They may decide to provide an extended review of many molecules on surfaces or alternatively keep a tutorial-like review as it is being published by RSC. Some details on the tutorial-like review article are provided at the end of this evaluation.

We thank the reviewer for the very important comments.

We have added a more detailed description and we have rewritten some paragraphes  in the manuscript. Thanks !

Q1: a) Please add molecular formulas of each molecule you describe on surfaces.

R1 : We thank for this comment. We have added the molecular formulas of each molecule.

Q2: b) for every molecule, please discuss its electronic structure (as an isolated molecule) and show its frontier orbitals

R2: Following the referee’s comments, we have included the models of the individual molecules (Fig. 1), the frontier orbitals HOMO/LUMO (Fig 2 and 3), and more sentences, lines  130-138 :

"To describe why the experimental and calculated STM images of these Lander molecules show that the functional groups, DCI and DAT, are not visible, we performed molecular frontier orbitals, of Lander-DAT and Lander-DCI alone (Figure 1 ((g), (h)) and Figure 2 ((d), (e))), by using the EHMO method as implemented in the YAeHMOP (Yet Another Extended Hückel Molecular Orbital Package) code [42-44], in these tunnelling conditions, we show a contribution from the HOMO frontier orbital in the experimental and calculated STM images of both Landers, because the bias voltage (V) used in all cases is lower than the highest occupied molecular orbital (HOMO)-lowest unoccupied molecular orbital (LUMO) gap (of approximatively, 2.65 V for Lander-DAT and 2.05 V for Lander-DCI) [7,29,45,46]"

Q3 : c) Provide a section describing the typical software and alternatives for calculating the STM images

R3 : We agree with this comment and we have added the following to the methods section (lines : 90-97) :

“this technique, based on the EHMO (Extended Hückel molecular orbital) method, offers a means of studying the transmission of electrons through a defect embedded in an infinite, periodic chain. In 1988 [38-40], the method was applied to the study of the transmission of electrons through a molecular switch embedded in a conducting polymer. More recently [41], the method was further developed so that it could allow the study of tunneling of electrons in a Scanning Tunnelling Microscope (STM) consisting of the apex, of the molecule being imaged, and the substrate, where the molecule is adsorbed. These results of simulations of STM images show...."

Q4: d) For isolated models, provide LDOS and band structure.

R4: We believe that the frontier orbitals of the molecules included in Figures 2 and 3 are sufficient to explain the results of the STM images. Please refer to our response to Comment b.

Q5 : e) If you aim a tutorial review, please also provide geometries and example input/output of the calculated structures/STM images.

R 5 : We can not insert the input and output files, because they are large, but we have added an image of the system (substrate, molecule, tip apex) in Figure 1.

Q 6 : f) Minor usage of figures from other papers is fine, but for review, please produce your own figures and be consistent with the colouring.

R 6 : We have produced our figures, which are clean and consistent with the coloring.

Comments on the Quality of English Language

Compared to the other aspects, there are no major concerns about the English language at this stage. Many sentences may appear grammatically correct,

Once again, we are grateful to the reviewer for the positive remarks on our work.

but they imply new semantics for common words.

An example is the word methodology. Both STM and DFT would be some form of developed methods (methodologies maybe if still in development). The "authors use these methodologies to reveal some new methodologies in the area of self-assembly"  (see copied sections below).

This does not make any sense, as the characterisation and description methods are used to characterise/describe the outcomes of molecular engineering but not the methods of molecular engineering per se.

” Herein, we review a series of studies 9 utilizing combining scanning tunneling microscopy (STM) with theoretical calculations, to reveal the methodologies used in the area of self-assembled driven by molecule Landers equipped with func- 11 tional groups on the surface.   The methodologies adopted here 232 are expected to be useful for researchers to advance the self-assembly of Lander molecules 233 directed by non-covalent interactions study in the future, within the emerging field of 234 nanotechnology and nanodevices”

R7 : We agree with the referre and we have changed the word methodology to the word process.

end of point-to-point response.

With best regard.

Round 2

Reviewer 1 Report

Comments and Suggestions for Authors

The authors have provided a revised form of their manuscript, answering to my comments and largely improving their old version.

Nothing to say, my recommendation is to accept the manuscript in present form

Author Response

Reviewer: 1

Comments: «The authors have provided a revised form of their manuscript, answering to my comments and largely improving their old version.

Nothing to say, my recommendation is to accept the manuscript in present form»

Answer: Once again, we are grateful to the reviewer for the positive remarks on our work.

Reviewer 2 Report

Comments and Suggestions for Authors

The authors have partially followed my recommendations. As the iThenticate report is still very alarming, I cannot support the paper publication at this stage, even under minor revision needed.

My recommendation is that the editor also share the iThenticate report with the authors.

Authors go carefully through the sections especially those that require major rewriting such as Abstract and Introduction, Adsorption of Single Molecule Landers on Metal Surfaces, and both sections on Self-Assembly Formed by Lander Molecules.

The authors responded to my follow-up comments and amended their article.

Considering the narrow scope of the topic, I suggest this article is considered as a minireview.

-----

Responses to the author's comments.

####################

Comment 1 : Over the past two decades the number of molecules deposited on surfaces and described using computational methods (including simulated STM) is on the scale of thousands. Molecules are being deposited for various reasons (e.g. qubits, memristors, catalysis etc). Many of them may be isolated or arrange themselves in some new orientation with respect to the surface. In this regard, the potential scope of the review would be significantly larger. In the present case, the authors have provided a review article with barely 40 citations. Although MDPI may not have strictly specified the number of citations involved with a typical review article, a "Minireview" in chemistry, as published by Wiley, is typically 5000 words, including around 100 references. The same is true for Nature Chemistry Reviews.

Answer 1: As for the H-bonding and electrostatic interactions directed supramolecular nanostructures, there are too many publications to be cited all.

Even so, we still add a few references as requested by this referee.

Gourdon, A. Synthesis of “molecular Landers”. European journal of organic chemistry 1998, 12, 2797–2801.

Rosei, F.; Schunack, M.; Naitoh, Y.; Jiang, P.; Gourdon, A.; Laegsgaard, E.; Stensgaard, I.; Joachim, C.; Besenbacher, F. Properties of large organic molecules on metal surfaces. Progress in Surface Science 2003 71, 95–146.

Aviram, A.; Ratner, M. A.; Molecular rectifiers. Chemical Physics Letters 1974, 29, 277-–283.

Sautet, P.; Joachim, C.; Electronic interference produced by a benzene embedded in a polyacetylene chain. Chemical Physics Letters 1988, 153, 511—516.

Sautet, P.; Joachim, C. The switching ability of a three-level tight-binding system: the isolated and embedded case. The Journal of Chemical Physics 1988, 21, 3939-–3957.

Sautet, P.; Joachim, C. Electronic transmission coefficient for the single-impurity problem in the scattering-matrix approach. Phys. Rev. B 1988, 38, 12238—12247.

Landrum, Greg, and Wingfield Glassey. "Yet Another Extended Hückel Molecular Orbital Package (YAeHMOP) Version 3.0 User Manual." (2006).

Coratger, R., Calmettes, B., Benjalal, Y., Bouju, X., & Coudret, C. (2012). Structural and electronic properties of hexa-adamantyl-hexa-phenylbenzene molecules studied by low temperature scanning tunneling microscopy. Surface science, 606(3-4), 444-449.

Ge, X.; Kuntze, J.; Berndt, R.; Tang, H.; Gourdon, A. Tunneling spectroscopy of lander molecules on coinage metal surfaces. Chemical Physics Letters 2008, 458, 161–165.

Ruiz, E.; Alvarez, S.; Hoffmann, R.; Bernstein, J. J. Am. Chem. Soc. 1994, 8207--8221.

Hughbanks, T.; Hoffmann, R. Chains of trans-edge-sharing molybdenum octahedra: metal-metal bonding in extended systems. Journal of the American Chemical Society 1983, 105, 3528-3537.

==This is not want I wanted

This is what is called ‘dumping of references’. In order to reach a few more references, the authors duly cite the work without properly reviewing it. That is not acceptable. Why are those references included and not others? What is their significance? Please write and elaborate.

####################

Comment 2 : In some cases, I can understand that originally, the authors have not clearly explained how their article is being positioned in the overall literature. However, after carefully re-reading the article, I see issues with its depth. For theoretical work, the authors would be expected to explain the molecular aspects of the surface chemistry or the overall theoretical methodologies. A current perspective by Duan and Xu shows how this is properly done https://doi.org/10.1021/acs.jpclett.3c01603. Although their article is not extensive (60 references) , it provides depth in understanding the role of the tip and substrate on the accuracy of the calcalculation. In the present submission, the authors do not go much in depth. There is no explanation of the molecular structure and properties of the involved molecules, no discussion on the computational methods, their limits and advantages for addressing the particular problem.

In addition to the overall formatting, the 55% iThenticate report is highly concerning. It appears as if the authors did very little in terms of their own reviewing (many phrases are completely borrowed from abstracts/conclusion statements of published papers).

Owing to these major reasons, I cannot recommend the article for publication in the present case. I suggest the authors undergo a major revision. They may decide to provide an extended review of many molecules on surfaces or alternatively keep a tutorial-like review as it is being published by RSC. Some details on the tutorial-like review article are provided at the end of this evaluation.

We thank the reviewer for the very important comments.

We have added a more detailed description and we have rewritten some paragraphes in the manuscript. Thanks !

==The iThenticate report is still  50%! Not much was changed obviously

####################

Q1: a) Please add molecular formulas of each molecule you describe on surfaces.

R1 : We thank for this comment. We have added the molecular formulas of each molecule.

====molecular structural formula – draw it in chemsketch / chem draw or any other chemical editor and then included it. Empirical formulas as you have included are not useful.

####################

Q2: b) for every molecule, please discuss its electronic structure (as an isolated molecule) and show its frontier orbitals

R2: Following the referee’s comments, we have included the models of the individual molecules (Fig. 1), the frontier orbitals HOMO/LUMO (Fig 2 and 3), and more sentences, lines 130-138 :

"To describe why the experimental and calculated STM images of these Lander molecules show that the functional groups, DCI and DAT, are not visible, we performed molecular frontier orbitals, of Lander-DAT and Lander-DCI alone (Figure 1 ((g), (h)) and Figure 2 ((d), (e))), by using the EHMO method as implemented in the YAeHMOP (Yet Another Extended Hückel Molecular Orbital Package) code [42-44], in these tunnelling conditions, we show a contribution from the HOMO frontier orbital in the experimental and calculated STM images of both Landers, because the bias voltage (V) used in all cases is lower than the highest occupied molecular orbital (HOMO)-lowest unoccupied molecular orbital (LUMO) gap (of approximatively, 2.65 V for Lander-DAT and 2.05 V for Lander-DCI) [7,29,45,46]"

==== This is fine , However, the HOMO/LUMO need to be in different lines so they appear larger.

 ####################

Q3 : c) Provide a section describing the typical software and alternatives for calculating the STM images

R3 : We agree with this comment and we have added the following to the methods section (lines : 90-97) :

“this technique, based on the EHMO (Extended Hückel molecular orbital) method, offers a means of studying the transmission of electrons through a defect embedded in an infinite, periodic chain. In 1988 [38-40], the method was applied to the study of the transmission of electrons through a molecular switch embedded in a conducting polymer. More recently [41], the method was further developed so that it could allow the study of tunneling of electrons in a Scanning Tunnelling Microscope (STM) consisting of the apex, of the molecule being imaged, and the substrate, where the molecule is adsorbed. These results of simulations of STM images show...."

==== THIS IS FINE ,

 ####################

Q4: d) For isolated models, provide LDOS and band structure.

R4: We believe that the frontier orbitals of the molecules included in Figures 2 and 3 are sufficient to explain the results of the STM images. Please refer to our response to Comment b.

==== FINE

 ####################

Q5 : e) If you aim a tutorial review, please also provide geometries and example input/output of the calculated structures/STM images.

R 5 : We can not insert the input and output files, because they are large, but we have added an image of the system (substrate, molecule, tip apex) in Figure 1.

==== I am not saying that you copy and paste the files inside the review. Just provide a link to a repository where they can be obtained and tested.  

 ####################

Q 6 : f) Minor usage of figures from other papers is fine, but for review, please produce your own figures and be consistent with the colouring.

R 6 : We have produced our figures, which are clean and consistent with the coloring.

==== OK good to confirm!

Comments on the Quality of English Language

Title Section

- Keep careful with the space "Self-assembly of molecular Landers equipped with functional moieties on the surface : a review." should be "Self-assembly of molecular Landers equipped with functional moieties on the surface: a review."

- Title – I am not in favour the word a review as judging by the style and the scope and encompassed literature, this is more of a minireview.

Abstract section (p. 1, lines 6-15):

-          Line 6: "which" should be "that."

-          Lines 7-9: "at surfaces" should be "on surfaces," and "allowing" should be "that allows."

Introduction section (p. 1-2, lines 16-47):

-          Line 16: "by" should be "of."

-          Lines 18-19: "the molecules organics" should be "organic molecules."

-          Line 21-22: "may indeed be possible" should be "is indeed possible."

-          Lines 32-33: Missing comma: "Au(111) highly oriented" should be "Au(111), highly oriented."

Self-assembly of Lander Molecules Guided by vdW and HB Interactions (p. 3-4, lines 119-138):

-          Lines 130-133: "we performed molecular frontier orbitals, of Lander-DAT and Lander-DCI alone" should be "we performed molecular frontier orbital calculations of Lander-DAT and Lander-DCI alone."

-          Lines 36-38: Missing period: "metal surface, as early as 2000," should be "metal surface. As early as 2000,"

Adsorption of Single Molecule Landers on Metal Surfaces section (p. 3, lines 64-138):

-          Line 65: Add "and intramolecular deformation" after "metallic nanostructures."

-          Line 68: "moulds" should be "molds."

-          Line 72: "by combination" should be "by combining."

-          Line 79: Remove "they have found."

-          Lines 80-84: Add "and" before "the two DAT groups," and replace "rotated, moreover" with "rotated. Moreover,"

-          Line 86: "interaction" should be "interactions."

-          Line 89: Replace comma with period after "[37]."

-          Lines 97-99: "lobe" should be "lobes," add period after "center."

-          Lines 102-103: "These" should be "This."

-          Lines 110-111: "smaller" should be "less corrugated."

-          Line 113: Remove "they have."

-          Self-Assembly Formed by Lander Molecules Guided by vdW and HB Interactions (p. 4, lines 139-193):

-          Line 141: Replace comma with period after "surface."

-          Line 143: Use single quotes and add commas: 'Four-Blade Mill,' 'Transition,' 'Stripe,'

-          Lines 147-148: Add comma after "groups."

-          Lines 149-152: Remove commas and restructure sentence for clarity.

-          Line 156: Add "the" before "2D network."

-          Line 160: Replace "by" with "through."

-          Lines 163-164: Remove comma after "investigated."

-          Lines 170-171: Replace "on the one hand," with "while."

-          Line 174: Remove comma after "heteromolecular."

-          Lines 177-179: Add commas around "DAT and DCI."

Self-Assembly Formed by Lander Molecules Guided by vdW, Hydrogen-Bonded, and Electrostatic Interactions (p. 5, lines 198-248):

-          Lines 230-231: "Distances averages" should be "average distances."

-          Lines 235-236: Remove space before colon.

Conclusions section (p. 6, lines 249-265):

-          Lines 252-254: Missing commas: "of the Lander facilitated by electrostatic attraction is significantly stronger" should be "of the Lander, facilitated by electrostatic attraction, is significantly stronger."

Author Response

Reviewer: 2

Comments:

The authors have partially followed my recommendations. As the iThenticate report is still very alarming, I cannot support the paper publication at this stage, even under minor revision needed.

My recommendation is that the editor also share the iThenticate report with the authors.

Authors go carefully through the sections especially those that require major rewriting such as Abstract and Introduction, Adsorption of Single Molecule Landers on Metal Surfaces, and both sections on Self-Assembly Formed by Lander Molecules.

The authors responded to my follow-up comments and amended their article.

Considering the narrow scope of the topic, I suggest this article is considered as a minireview.

Answer: We have revised the title section to be: 

“Self-assembly of molecular Landers equipped with functional moieties on the surface: a mini review 

-----

Responses to the author's comments.

####################

Comment 1 : Over the past two decades the number of molecules deposited on surfaces and described using computational methods (including simulated STM) is on the scale of thousands. Molecules are being deposited for various reasons (e.g. qubits, memristors, catalysis etc). Many of them may be isolated or arrange themselves in some new orientation with respect to the surface. In this regard, the potential scope of the review would be significantly larger. In the present case, the authors have provided a review article with barely 40 citations. Although MDPI may not have strictly specified the number of citations involved with a typical review article, a "Minireview" in chemistry, as published by Wiley, is typically 5000 words, including around 100 references. The same is true for Nature Chemistry Reviews.

Answer 1: As for the H-bonding and electrostatic interactions directed supramolecular nanostructures, there are too many publications to be cited all.

Even so, we still add a few references as requested by this referee.

Gourdon, A. Synthesis of “molecular Landers”. European journal of organic chemistry 1998, 12, 2797–2801.

Rosei, F.; Schunack, M.; Naitoh, Y.; Jiang, P.; Gourdon, A.; Laegsgaard, E.; Stensgaard, I.; Joachim, C.; Besenbacher, F. Properties of large organic molecules on metal surfaces. Progress in Surface Science 2003 71, 95–146.

Aviram, A.; Ratner, M. A.; Molecular rectifiers. Chemical Physics Letters 1974, 29, 277-–283.

Sautet, P.; Joachim, C.; Electronic interference produced by a benzene embedded in a polyacetylene chain. Chemical Physics Letters 1988, 153, 511—516.

Sautet, P.; Joachim, C. The switching ability of a three-level tight-binding system: the isolated and embedded case. The Journal of Chemical Physics 1988, 21, 3939-–3957.

Sautet, P.; Joachim, C. Electronic transmission coefficient for the single-impurity problem in the scattering-matrix approach. Phys. Rev. B 1988, 38, 12238—12247.

Landrum, Greg, and Wingfield Glassey. "Yet Another Extended Hückel Molecular Orbital Package (YAeHMOP) Version 3.0 User Manual." (2006).

Coratger, R., Calmettes, B., Benjalal, Y., Bouju, X., & Coudret, C. (2012). Structural and electronic properties of hexa-adamantyl-hexa-phenylbenzene molecules studied by low temperature scanning tunneling microscopy. Surface science, 606(3-4), 444-449.

Ge, X.; Kuntze, J.; Berndt, R.; Tang, H.; Gourdon, A. Tunneling spectroscopy of lander molecules on coinage metal surfaces. Chemical Physics Letters 2008, 458, 161–165.

Ruiz, E.; Alvarez, S.; Hoffmann, R.; Bernstein, J. J. Am. Chem. Soc. 1994, 8207--8221.

Hughbanks, T.; Hoffmann, R. Chains of trans-edge-sharing molybdenum octahedra: metal-metal bonding in extended systems. Journal of the American Chemical Society 1983, 105, 3528-3537.

==This is not want I wanted

This is what is called ‘dumping of references’. In order to reach a few more references, the authors duly cite the work without properly reviewing it. That is not acceptable. Why are those references included and not others? What is their significance? Please write and elaborate.

We have added these references for

Answer: We have add references to describe Landers molecules, to clarify methods and calculation codes such as Yaehmop, ESQC,  and also for non-covalent interactions...

Landers Molecules  :

Gourdon, A. Synthesis of “molecular Landers”. European journal of organic chemistry 1998, 12, 2797–2801.   (the previous version)

Rosei, F.; Schunack, M.; Naitoh, Y.; Jiang, P.; Gourdon, A.; Laegsgaard, E.; Stensgaard, I.; Joachim, C.; Besenbacher, F. Properties of large organic molecules on metal surfaces. Progress in Surface Science 2003 71, 95–146.(the previous version)

Ge, X.; Kuntze, J.; Berndt, R.; Tang, H.; Gourdon, A. Tunneling spectroscopy of lander molecules on coinage metal surfaces. Chemical Physics Letters 2008, 458, 161–165.(the previous version)

ESQC technique  :

Aviram, A.; Ratner, M. A.; Molecular rectifiers. Chemical Physics Letters 1974, 29, 277-–283. (the previous version)

Sautet, P.; Joachim, C.; Electronic interference produced by a benzene embedded in a polyacetylene chain. Chemical Physics Letters 1988, 153, 511—516. (the previous version)

Sautet, P.; Joachim, C. The switching ability of a three-level tight-binding system: the isolated and embedded case. The Journal of Chemical Physics 1988, 21, 3939-–3957. (the previous version)

Sautet, P.; Joachim, C. Electronic transmission coefficient for the single-impurity problem in the scattering-matrix approach. Phys. Rev. B 1988, 38, 12238—12247. (the previous version)

Coratger, R., Calmettes, B., Benjalal, Y., Bouju, X., & Coudret, C. (2012). Structural and electronic properties of hexa-adamantyl-hexa-phenylbenzene molecules studied by low temperature scanning tunneling microscopy. Surface science, 606(3-4), 444-449. (the previous version)

YAeHMOP code

Landrum, Greg, and Wingfield Glassey. "Yet Another Extended Hückel Molecular Orbital Package (YAeHMOP) Version 3.0 User Manual." (2006). (the previous version)

Ruiz, E.; Alvarez, S.; Hoffmann, R.; Bernstein, J. J. Am. Chem. Soc. 1994, 8207--8221. (the previous version)

https://yaehmop.sourceforge.net (this version)

Non-covalent interactions (coordination bonds and electrostatic interactions)

Dmitriev, A.; Spillmann, H.; Lin, N.; Barth, J. V.; Kern, K. Modular assembly of two-dimensional metal–organic coordination 347 networks at a metal surface. Angewandte Chemie International Edition, 2003, 42, 2670–2673. (this version)

Shi, Z.; Lin, N. Structural and chemical control in assembly of multicomponent metalorganic coordination networks on a surface. 345 Journal of the American Chemical Society 2010, 31, 10756–10761 (this version)

Alvarez, L.; Peláez, S.; Caillard, R.; Serena, P. A.; Martín-Gago, J. A.; Méndez, J. Metal-organic extended 2D structures: Fe-PTCDA 343 on Au (111). Nanotechnology, 2010 21, 305703, 1–7. (this version)

Liu, J.; Lin, N. On-Surface-Assembled Single-Layer Metal-Organic Frameworks with Extended Conjugation. ChemPlusChem 2023, 341 88, 2–9 (this version)

Jensen, S.; Baddeley, C. J. Formation of PTCDI-based metal-organic structures on a Au (111) surface modified by 2-D Ni clusters. 349 The Journal of Physical Chemistry C 2008, 112, 15439–15448. (this version)

####################

Comment 2 : In some cases, I can understand that originally, the authors have not clearly explained how their article is being positioned in the overall literature. However, after carefully re-reading the article, I see issues with its depth. For theoretical work, the authors would be expected to explain the molecular aspects of the surface chemistry or the overall theoretical methodologies. A current perspective by Duan and Xu shows how this is properly done https://doi.org/10.1021/acs.jpclett.3c01603. Although their article is not extensive (60 references) , it provides depth in understanding the role of the tip and substrate on the accuracy of the calcalculation. In the present submission, the authors do not go much in depth. There is no explanation of the molecular structure and properties of the involved molecules, no discussion on the computational methods, their limits and advantages for addressing the particular problem.

In addition to the overall formatting, the 55% iThenticate report is highly concerning. It appears as if the authors did very little in terms of their own reviewing (many phrases are completely borrowed from abstracts/conclusion statements of published papers).

Owing to these major reasons, I cannot recommend the article for publication in the present case. I suggest the authors undergo a major revision. They may decide to provide an extended review of many molecules on surfaces or alternatively keep a tutorial-like review as it is being published by RSC. Some details on the tutorial-like review article are provided at the end of this evaluation.

We thank the reviewer for the very important comments.

Answer: Sorry, the first time, we didn't receive the report!

We have added a more detailed description and we have rewritten some paragraphes in the manuscript. Thanks !

==The iThenticate report is still  50%! Not much was changed obviously

Answer: This time we received it, and to decrease the report, we have rewritten and thoroughly revised all paragraphs in the manuscript. Thanks!

####################

Q1: a) Please add molecular formulas of each molecule you describe on surfaces.

R1 : We thank for this comment. We have added the molecular formulas of each molecule.

====molecular structural formula – draw it in chemsketch / chem draw or any other chemical editor and then included it. Empirical formulas as you have included are not useful.

Answer: Following the referee’s comments, we have added the chemical structure of each Lander in figures 2(a) and 4(a) of the individual molecules.

####################

Q2: b) for every molecule, please discuss its electronic structure (as an isolated molecule) and show its frontier orbitals

R2: Following the referee’s comments, we have included the models of the individual molecules (Fig. 1), the frontier orbitals HOMO/LUMO (Fig 2 and 3), and more sentences, lines 130-138 :

"To describe why the experimental and calculated STM images of these Lander molecules show that the functional groups, DCI and DAT, are not visible, we performed molecular frontier orbitals, of Lander-DAT and Lander-DCI alone (Figure 1 ((g), (h)) and Figure 2 ((d), (e))), by using the EHMO method as implemented in the YAeHMOP (Yet Another Extended Hückel Molecular Orbital Package) code [42-44], in these tunnelling conditions, we show a contribution from the HOMO frontier orbital in the experimental and calculated STM images of both Landers, because the bias voltage (V) used in all cases is lower than the highest occupied molecular orbital (HOMO)-lowest unoccupied molecular orbital (LUMO) gap (of approximatively, 2.65 V for Lander-DAT and 2.05 V for Lander-DCI) [7,29,45,46]"

==== This is fine , However, the HOMO/LUMO need to be in different lines so they appear larger.

Answer: We have included the FMOs (frontier molecular orbitales) of the individual molecule (initially presented in Figures 2 and 3) as Figures 3 and 5.

 ####################

Q3 : c) Provide a section describing the typical software and alternatives for calculating the STM images

R3 : We agree with this comment and we have added the following to the methods section (lines : 90-97) :

“this technique, based on the EHMO (Extended Hückel molecular orbital) method, offers a means of studying the transmission of electrons through a defect embedded in an infinite, periodic chain. In 1988 [38-40], the method was applied to the study of the transmission of electrons through a molecular switch embedded in a conducting polymer. More recently [41], the method was further developed so that it could allow the study of tunneling of electrons in a Scanning Tunnelling Microscope (STM) consisting of the apex, of the molecule being imaged, and the substrate, where the molecule is adsorbed. These results of simulations of STM images show...."

==== THIS IS FINE ,

Answer: We thank for this comment.

 ####################

Q4: d) For isolated models, provide LDOS and band structure.

R4: We believe that the frontier orbitals of the molecules included in Figures 2 and 3 are sufficient to explain the results of the STM images. Please refer to our response to Comment b.

==== FINE

Answer: We thank for this comment.

 ####################

Q5 : e) If you aim a tutorial review, please also provide geometries and example input/output of the calculated structures/STM images.

R 5 : We can not insert the input and output files, because they are large, but we have added an image of the system (substrate, molecule, tip apex) in Figure 1.

==== I am not saying that you copy and paste the files inside the review. Just provide a link to a repository where they can be obtained and tested.  

For Mopac (free code for academic) we have added this reference http://openmopac.net/ (examples of input/output files) in bibliography.

YaeHMOP (free code) : https://yaehmop.sourceforge.net/ (examples of input/output files)

But for ESQC and MM4 we cannot provide a link because these codes are not free.

And we added these sections in the review

«Data Availability Statement: The input/output files for ESQC, MM4, Mopac, and YAeHMOP codes, used in this study for theoretical calculations will be made available by the authors on request (only for free codes).»

####################

Q 6 : f) Minor usage of figures from other papers is fine, but for review, please produce your own figures and be consistent with the colouring.

R 6 : We have produced our figures, which are clean and consistent with the coloring.

==== OK good to confirm!

Answer: We thank for this comment.

Title Section

- Keep careful with the space "Self-assembly of molecular Landers equipped with functional moieties on the surface : a review." should be "Self-assembly of molecular Landers equipped with functional moieties on the surface: a review."

- Title – I am not in favour the word a review as judging by the style and the scope and encompassed literature, this is more of a minireview.

Abstract section (p. 1, lines 6-15):

-          Line 6: "which" should be "that."

-          Lines 7-9: "at surfaces" should be "on surfaces," and "allowing" should be "that allows."

Introduction section (p. 1-2, lines 16-47):

-          Line 16: "by" should be "of."

-          Lines 18-19: "the molecules organics" should be "organic molecules."

-          Line 21-22: "may indeed be possible" should be "is indeed possible."

-          Lines 32-33: Missing comma: "Au(111) highly oriented" should be "Au(111), highly oriented."

Self-assembly of Lander Molecules Guided by vdW and HB Interactions (p. 3-4, lines 119-138):

-          Lines 130-133: "we performed molecular frontier orbitals, of Lander-DAT and Lander-DCI alone" should be "we performed molecular frontier orbital calculations of Lander-DAT and Lander-DCI alone."

-          Lines 36-38: Missing period: "metal surface, as early as 2000," should be "metal surface. As early as 2000,"

Adsorption of Single Molecule Landers on Metal Surfaces section (p. 3, lines 64-138):

-          Line 65: Add "and intramolecular deformation" after "metallic nanostructures."

-          Line 68: "moulds" should be "molds."

-          Line 72: "by combination" should be "by combining."

-          Line 79: Remove "they have found."

-          Lines 80-84: Add "and" before "the two DAT groups," and replace "rotated, moreover" with "rotated. Moreover,"

-          Line 86: "interaction" should be "interactions."

-          Line 89: Replace comma with period after "[37]."

-          Lines 97-99: "lobe" should be "lobes," add period after "center."

-          Lines 102-103: "These" should be "This."

-          Lines 110-111: "smaller" should be "less corrugated."

-          Line 113: Remove "they have."

-          Self-Assembly Formed by Lander Molecules Guided by vdW and HB Interactions (p. 4, lines 139-193):

-          Line 141: Replace comma with period after "surface."

-          Line 143: Use single quotes and add commas: 'Four-Blade Mill,' 'Transition,' 'Stripe,'

-          Lines 147-148: Add comma after "groups."

-          Lines 149-152: Remove commas and restructure sentence for clarity.

-          Line 156: Add "the" before "2D network."

-          Line 160: Replace "by" with "through."

-          Lines 163-164: Remove comma after "investigated."

-          Lines 170-171: Replace "on the one hand," with "while."

-          Line 174: Remove comma after "heteromolecular."

-          Lines 177-179: Add commas around "DAT and DCI."

Self-Assembly Formed by Lander Molecules Guided by vdW, Hydrogen-Bonded, and Electrostatic Interactions (p. 5, lines 198-248):

-          Lines 230-231: "Distances averages" should be "average distances."

-          Lines 235-236: Remove space before colon.

Conclusions section (p. 6, lines 249-265):

-          Lines 252-254: Missing commas: "of the Lander facilitated by electrostatic attraction is significantly stronger" should be "of the Lander, facilitated by electrostatic attraction, is significantly stronger."

Answer: We thank the reviewer again for the great effort and have made the revisions accordingly in the revised manuscript.  

Round 3

Reviewer 2 Report

Comments and Suggestions for Authors I understand that I have been a little bit critical to the previous submissions, however, I acknowledge that the authors have placed an effort and have prepared new submission that is far more presentable than the previous iterations. However, there are a number of points that the authors can change before the paper is considered for publication: Minor / formatting points: a) The authors are expected to provide doi links to their references. b) When reusing with permissions from previous works the authors need to specify to which subfigure these permissions apply. c) When using a subfigure such as Figure 8a that comes from reference [31] the authors need to be careful how they document the permission. The particular figure is a composite one where the bottom layer comes from experiment, while on the top layer there are images of molecules. The figure published in reference 31 contains the same figure components, however here the molecules are being shifted for clarity. This is a modification that renders the figure technically different than what has been published. Thus maybe the formulation “redrawn” from reference 31 would apply better. Other issues: d) I have asked the authors in this theoretical paper to discuss the electronic structure of the free molecules and possibly to discuss how the adsorption on the surfaces influences it. When they do this, there will be new original thoughts and thus will didactical value and improve on the concerning iThenticate report. The authors have added Figure 3 and Figure 5 in the paper, but they have not cited and discussed these figures in the text! That is not right. e) Ideally Figure 3 and 5 can obtain some colouring. For example, it is difficult to distinguish nitrogen from carbon atoms. Also the orbital lobes could be possibly show in different colouring. f) Although I have not seen the last iThenticate report, I obtained information from the Assistant Editor, Ms Stankovic that it is 31%. That is an improvement. The authors should go one more time carefully and understand where this percentage comes from. Considering that this is a minireview, even a small similarity will show higher percentage. If the similarities are in the references these sections can be ignored. However, the authors need to ensure that there are sentences with major overlaps. g) For people who typically work with DFT methods it is a little bit puzzling why one would resort to the Huckel method, considering the lower accuracy, especially nowadays when computational power is ubiquitous. The author should provide a comment on that. h) The key code for the EHMO-ESQC is basically used only by a single group in France who does not share it (collaborator of the corresponding author). Following this it is clear why the authors do not provide input / output files, as there is no didactical aspect when the code is kept privately. This is not necessarily helpful for the broader community. i) For the generation of STM images following VASP calculations there is "HIVE-STM" software that is open and free to use. The authors can make comparison to what they have. Can the computational STM images based on EHMO-ESQC be reproduced by HIVE-STM https://dannyvanpoucke.be/hive-stm-en/ ? please comment. Comments on the Quality of English Language Page 1, Row 25:
It is written: "design of nanostructures that could satisfy the requirements of nanoengineering concepts." It should be: "design of nanostructures that can satisfy the requirements of nanoengineering concepts."

Page 2, Row 17:
It is written: "This active domain has been widely studied several times." It should be: "This active domain has been widely studied." (remove – several times).

Page 3, Row 25:
It is written: "self-assembly of Lander molecules adsorbed on metallic substrates for forming distinct networks." It should be: "self-assembly of Lander molecules adsorbed on metallic substrates to form distinct networks."

Page 4, Row 50:
It is written: "The self-assembled of Lander molecules, by non-covalent interactions …" It should be: "The self-assembly of Lander molecules, driven by non-covalent interactions

Page 5, Row 12:
It is written: "functional moieties on the metallic surfaces, were synthesized and characterized by a combination of UHV-STM (Ultra-high vacuum-scanning tunneling microscopy) and theoretical studies." It should be: "functional moieties on the metallic surfaces were characterized by a combination of UHV-STM (Ultra-high vacuum-scanning tunneling microscopy) and described through theoretical studies."

Page 10, Row 237:
It is written: "SCF-PM6: Self-Consistent Field Parametric Method 6." It should be: "SCF-PM6: Self-Consistent Field Parameterization Method 6."

Author Response

More detailed comments and a commentary of the manuscript changes as following: 

Reviewer: 2 

Comments: 

I understand that I have been a little bit critical to the previous submissions, however, I acknowledge that the authors have placed an effort and have prepared new submission that is far more presentable than the previous iterations. However, there are a number of points that the authors can change before the paper is considered for publication: Minor / formatting points:  

The referee makes valuable and insightful suggestions for further improving the paper, every time we have addressed all these as detailed in our point-by-point response below. 

Answer: 

----- 

  1. a) The authors are expected to provide doi links to their references. 

We have provided doi links for the references.   

  1. b) When reusing with permissions from previous works the authors need to specify to which subfigure these permissions apply. 

We will add a more detailed description in the Figures regarding this question. Thanks! 

“Figure 2. (a) Chemical structure of Lander-DAT molecule (C64H68N10). (b) and (c) High-resolution STM images of a single Lander-DAT molecule on Cu(110) and Au(111) surfaces, respectively (It = 0.66 nA, Vt = 1.73 mV). (d) and (e) Space-filling models of Lander-DAT adsorbed on Cu(110) and Au(111) respectively, where N, H, C, and surface atoms are represented in blue, white, grey, and yellow respectively. (f) and (g) EHMO-ESQC calculated images of a Lander-DAT molecule adsorbed on Cu(110) and Au(111), respectively. ((b) and (f) are adapted and reprinted with permission from [9]. Copyright 2010 American Chemical Society. (a), (c), and (g) are adapted and reprinted with permission from [10]. Copyright 2009 Tsinghua University Press and Springer-Verlag Berlin Heidelberg)” 

“Figure 4. (a) Chemical structure of molecular Lander-DCI (C112H102N2O4 ). (b) STM image of an individual Lander-DCI on Au(111) substrate (It = 0.26 nA; Vt = 1239 mV). (c) Top view of the optimized chemical structure of a Lander-DCI on Au(111), where Au, Ni, H, O, N, and C atoms are in yellow, green, white, red, blue, and grey, respectively. (d) ESQC image of a Lander-DCI on Au(111) as shown in panel (c). ((a), (b), (c), and (d) are adapted and reprinted with permission from [30]. Copyright 2010 The Royal Society of Chemistry)” 

“Figure 6. (a) STM image of 1D chain of Lander-DAT on Au(111) surface (It = 0.70 nA, Vt = 1250  

mV). (b), (c), and (d) STM images of the ’Stripe’ structure, ’Four-Blade Mill’ structure, and ’Transition’ structure, respectively, on Au(111) (It = 0.48 nA, Vt = 1250 mV). (e), (f), (g), and (h) Optimized models for the 1D chain, ’Stripe’ structure, ’Four-Blade Mill’ structure, and ’Transition’ structure, respectively, where atoms of C, N, and H are in grey, blue, and white. (i), (j), (k), and (l) EHMO-ESQC calculated images of the 1D chain (i), ’Stripe’ structure (j), ’Four-Blade Mill’ structure (k), and ’Transition’ structure (l) as in experiments ((a)-(d)). ((a), (b), (c), (d), and (e) are adapted and reprinted with permission from [10]. Copyright 2010 American Chemical Society)” 

Figure 7. (a) STM image of the 1D chains of Lander-DCI on Au(111) terrace (It = 0.55 nA, Vt = 1250 mV). (b) and (c) STM images of a 1D chain of Lander-DCI on a terrace and a step edge of the Au(111) substrate, respectively. (d) Model of the 1D chain of Lander-DCI on Au(111), obtained from MM4 force field calculations. (e) EHMO-ESQC calculated image of a 1D chain of Lander-DCI on Au(111) as shown in panel (d). (f) STM image of the 2D network of Lander-DCI and the superimposed calculated model. ((a), (b), (c), (d), and (f) are adapted and reprinted with permission from [30]. Copyright 2010 The Royal Society of Chemistry)” 

“Figure 8. (a) STM image of the 2D assembly of DAT-DCI molecules on the Au(111) substrate and the superimposed calculated model. (b) Calculated model of the structure formed by Lander-DAT and Lander-DCI on Au(111), where Au, Ni, H, O, N, and C atoms are in yellow, green, white, red, blue, and grey, respectively. (c) Close-view of the three-dimensional hydrogen bonding between DCI and DAT groups. ((a) and (b) are adapted and reprinted with permission from [31]. Copyright 2014 The Royal Society of Chemistry)” 

“Figure 9. (a) STM image of 1D formed by PTCDI molecules and Ni atoms on Au(111) (It = 0.53  

nA,Vt = 1.20 V). (b) STM image of the ’0D’ nanostructure (It = 0.55 nA, Vt = 1.05 V). (c) STM image of ’Y-shape’ nanostructure (It = 0.44 nA, Vt = 0.71 V). (d) STM image of a 2D network of PTCDI-Ni (It = 0.50 nA, Vt = 1.75 V). (e-h) Calculated structures for the 1D chain, ’0D’, ’Y-shape’ nanostructures, and 2D network, where Au, Ni, H, O, N, and C atoms are in yellow, green, white, red, blue, and grey, respectively. (i-k) EHMO-ESQC calculated images of the 1D chain (i), ’0D’ cluster (j), and ’Y-shape’ nanostructure (k). ((a)-(h) are adapted and reprinted with permission from ref [53]. Copyright 2012 Tsinghua University Press and Springer-Verlag Berlin Heidelberg)” 

“Figure 10. (a) STM image of the 2D assembly formed by Lander-DAT and PTCDI-Ni on Au(111). 
(b) Calculated model of the 2D assembly (Lander-DAT and PTCDI-Ni) on Au(111) surface, where 
Au, Ni, H, O, N, and C atoms are in yellow, green, white, red, blue, and grey, respectively. (c) and 
(d) Close-view and Top-view of the 3D HB between CDI and DAT functional groups. ((a), (b), (c), 
and (d) are adapted and reprinted with permission from [ 55 ]. Copyright 2018 The Royal Society of Chemistry)” 

  1. c) When using a subfigure such as Figure 8a that comes from reference [31] the authors need to be careful how they document the permission. The particular figure is a composite one where the bottom layer comes from experiment, while on the top layer there are images of molecules. The figure published in reference 31 contains the same figure components, however here the molecules are being shifted for clarity. This is a modification that renders the figure technically different than what has been published. Thus maybe the formulation “redrawn” from reference 31 would apply better. Other issues: 

Yes, you are right we did not pay attention, the images are different. As part of the permission given, we have changed the image in Figure 8a to the image published in reference 31. 

  1. d) I have asked the authors in this theoretical paper to discuss the electronic structure of the free molecules and possibly to discuss how the adsorption on the surfaces influences it. When they do this, there will be new original thoughts and thus will didactical value and improve on the concerning iThenticate The authors have added Figure 3 and Figure 5 in the paper, but they have not cited and discussed these figures in the text! That is not right.

We have cited and discussed these figures in paragraph 2, page 4, lines 119-122. 

But for more detail, we have added more sentences:   

Lines 122-134 

These calculations, show that the electron density of the frontier molecular orbitals of DCI and DAT Landers is delocalized on the molecule, but for the LUMO orbital it is higher on both functional groups (DAT and DCI), on the core, and an average electron density over the spacer groups (legs), while the HOMO orbitals are characterized by a very high electron density on the legs (tert-butyl and bulky 3,5-di-tert-butylphenyl) and very low density on the functional moieties for both molecules. In addition, the Lander molecules are physisorbed on the metallic surfaces and the molecular board of each molecule Lander is elevated from the surface by the legs. Therefore, the surface does not influence the electronic structures of the molecules when they are adsorbed. From these results, we show that the morphology of the STM image comes from a contribution of the HOMO molecular orbital in the measurement and calculated images of both Landers, because the bias voltage (V) used in all cases is lower than the HOMO-LUMO gap of the DAT and DCI molecules (of approximatively, 2.65 V for Lander-DAT and 2.05 V for Lander-DCI) [9,29,46,47]. 

  1. e) Ideally Figure 3 and 5 can obtain some colouring. For example, it is difficult to distinguish nitrogen from carbon atoms.

We agree with the referee, there are several codes to calculate the shapes of molecular orbitals with different methods, such as Mopac, Gamess, Gaussian ... , but we made sure to use the same EHMO method, used in ESQC-STM calculated images, to calculate the FMOs of the molecules, and compare them with STM and ESQC images.  

To distinguish nitrogen from carbon atoms, we have added the following sentences: 

“Figure 3. Calculated frontier molecular orbitals, HOMO and LUMO, of Lander-DAT molecule free, where nitrogen, carbon, and hydrogen, atoms are colored in grey, black, and light grey, respectively.” 

“Figure 5. Calculated frontier molecular orbitals, HOMO and LUMO, of Lander-DCI molecule free, where carbon, oxygen, nitrogen, and hydrogen atoms are colored in black, dark grey, grey, and light grey, respectively. 

Moreover, the structures of the molecules have already been shown in Figures 2 (d) and 4 (c), from these structures we can distinguish the different atoms.” 

Also the orbital lobes could be possibly show in different colouring. 

For this code, the orbital lobes are represented differently, by solid lines and dashed lines. 

  1. f) Although I have not seen the last iThenticate report, I obtained information from the Assistant Editor, Ms Stankovic that it is 31%. That is an improvement. The authors should go one more time carefully and understand where this percentage comes from. Considering that this is a minireview, even a small similarity will show higher percentage. If the similarities are in the references these sections can be ignored. However, the authors need to ensure that there are sentences with major overlaps.

We have ensured that there are no sentences with major overlaps because when the references, are ignored, we found 24 % of  iThenticate report. 

  1. g) For people who typically work with DFT methods it is a little bit puzzling why one would resort to the Huckel method, considering the lower accuracy, especially nowadays when computational power is ubiquitous. The author should provide a comment on that.

Yes, we agree with the referee, but the EHMO method has shown its good performance in the study of organic/organometallic molecules isolated or adsorbed on metal surfaces: 

  • Optimize molecule structures and calculate molecular orbital (MO) shapes for isolated molecules. 
  • Optimize molecule structures adsorbed on the surfaces. 
  • A combination with the ESQC method allows us to calculate STM images of molecule structures adsorbed on surfaces.  
  • It determines the shapes of molecular orbitals visualized by scanning tunneling microscopy (STM), by comparison with MO. 
  • It determines the morphology of the nanostructures and identifies the different parts formed by molecules assembled on the surface. 
  • This method is not expensive.  
  • The most important is that the results found by this semi-empirical method (EHMO), in most of our works, are in good agreement with the experimental results (STM), which allows us to explain qualitatively and quantitatively the self-assembly on solid surfaces. 

Benjalal, Y., Bonvoisin, J., & Bouju, X. (2019). Unraveling the molecular conformations of a single ruthenium complex adsorbed on the Ag (111) surface by calculations. Physical Chemistry Chemical Physics, 21(19), 10022-10027. 

Calmettes, B., Vernisse, L., Guillermet, O., Benjalal, Y., Bouju, X., Coudret, C., & Coratger, R. (2014). Observation and manipulation of hexa-adamantyl-hexa-benzocoronene molecules by low temperature scanning tunneling microscopy. Physical Chemistry Chemical Physics, 16(41), 22903-22912. 

Vernisse, L., Munery, S., Ratel-Ramond, N., Benjalal, Y., Guillermet, O., Bouju, X., ... & Bonvoisin, J. (2012). UHV-STM Investigations and Numerical Calculations of a Ruthenium β-Diketonato Complex with Protected Ethynyl Ligand:[Ru (dbm) 2 (acac-TIPSA)]. The Journal of Physical Chemistry C, 116(25), 13715-13721. 

Coratger, R., Calmettes, B., Benjalal, Y., Bouju, X., & Coudret, C. (2012). Structural and electronic properties of hexa-adamantyl-hexa-phenylbenzene molecules studied by low temperature scanning tunneling microscopy. Surface science, 606(3-4), 444-449. 

Munery, S., Ratel‐Ramond, N., Benjalal, Y., Vernisse, L., Guillermet, O., Bouju, X., ... & Bonvoisin, J. (2011). Synthesis and Characterization of a Series of Ruthenium Tris (β‐diketonato) Complexes by an UHV‐STM Investigation and Numerical Calculations. 

  1. i) For the generation of STM images following VASP calculations there is "HIVE-STM" software that is open and free to use. The authors can make comparison to what they have. Can the computational STM images based on EHMO-ESQC be reproduced by HIVE-STM https://dannyvanpoucke.be/hive-stm-en/ ? please comment.

Considering the type and quantity of scientific works produced by the "HIVE-STM" software, which I did not know frankly, it is a wonderful and effective program and can be used in an easy way to study some phenomena of self-assembly. 

As well as the “HIVE-STM” Software is open and free to use! 

But the question remains, can it be used to study large molecules  (for example, DCI 220 atoms, DAT 140 atoms, )? 

Because, you have seen that we studied self-assembled systems of DCI and DAT molecules ....   formed by 600 atoms (molecules) + 300 atoms (the surface) and 100 atoms (the tip), a total of 1000 atoms. 

And does it contain the parameters of most atoms (N, Ni, Cr, Ru ....)? 

If so, it can be relied upon, especially to produce images and characterize the adsorption of molecules on substrates. 

To know more, frankly, I will try to contact the developer (Dr. Danny E. P. Vanpouck) and why not use this program in future work. 

---------------------------------------------------------------------------------------------------------------------- 

Comments on the Quality of English Language 

Page 1, Row 25: 
It is written: "design of nanostructures that could satisfy the requirements of nanoengineering concepts." It should be: "design of nanostructures that can satisfy the requirements of nanoengineering concepts." 

Page 2, Row 17: 
It is written: "This active domain has been widely studied several times." It should be: "This active domain has been widely studied." (remove – several times). 

Page 3, Row 25: 
It is written: "self-assembly of Lander molecules adsorbed on metallic substrates for forming distinct networks." It should be: "self-assembly of Lander molecules adsorbed on metallic substrates to form distinct networks." 

Page 4, Row 50: 
It is written: "The self-assembled of Lander molecules, by non-covalent interactions …" It should be: "The self-assembly of Lander molecules, driven by non-covalent interactions 

Page 5, Row 12: 
It is written: "functional moieties on the metallic surfaces, were synthesized and characterized by a combination of UHV-STM (Ultra-high vacuum-scanning tunneling microscopy) and theoretical studies." It should be: "functional moieties on the metallic surfaces were characterized by a combination of UHV-STM (Ultra-high vacuum-scanning tunneling microscopy) and described through theoretical studies." 

Page 10, Row 237: 
It is written: "SCF-PM6: Self-Consistent Field Parametric Method 6." It should be: "SCF-PM6: Self-Consistent Field Parameterization Method 6." 

Answer: We thank the reviewer again and have made the revisions accordingly in this revised manuscript.  

Wa have added  

* Correspondence: y.benjalal@usms.ma 

Keywords: Self-assembly; molecular Landers; functional groups; scanning tunneling microscopy; van der Waals forces; hydrogen bonding; electrostatic interaction; theoretical calculations; frontier molecular orbitals 

Pr. Youness BENJALAL 

Round 4

Reviewer 2 Report

Comments and Suggestions for Authors

The authors have answered my questions. 

Comments on the Quality of English Language

Page 1, Line 22: "This active domain has been widely studied. [4–6]." -> "This active domain has been widely studied [4–6]." Pls format refernce

Page 1, Line 26-28: "...nanowires for potential use in emerging molecular electronic circuitry, utilizing the molding..." -> "...nanowires for potential use in emerging molecular electronic circuitry by utilizing the molding." Add  the word by and remove the comma

Page 1, Line 30: "...would be advisable. [9]." -> "...would be advisable [9]."  Pls format reference

Page 2, Line 55: " (HB, vdW force, electrostatic interaction, metal-organic coordination...)", -> instead of ... use "etc." 

Page 7, Line 103: "...guided by multiple non-covalent interactions on solid surfaces in the future." -> "...guided by multiple non-covalent interactions on solid surfaces."

Page 11, Line 217: "In this mini review, we have reviewed the self-assembled built from Lander molecules through intermolecular non-covalent interactions." -> "In this mini review, we have discussed the self-assembly of Lander molecules through intermolecular non-covalent interactions."

Page 11, Line 231: "...nanotechnology. 232" -> "...nanotechnology." Pls format refernce

Author Response

We thank the reviewer again.